# Advanced Medical Therapies in the Management of Non-Scarring Alopecia: Areata and Androgenic Alopecia

**DOI:** 10.3390/ijms21218390

**Published:** 2020-11-09

**Authors:** Antonio Martinez-Lopez, Trinidad Montero-Vilchez, Álvaro Sierra-Sánchez, Alejandro Molina-Leyva, Salvador Arias-Santiago

**Affiliations:** 1Dermatology Unit, Virgen de las Nieves University Hospital, Av de Madrid, 15, 18012 Granada, Spain; antoniomartinezlopez@aol.com (A.M.-L.); tmonterov@gmail.com (T.M.-V.); salvadorarias@hotmail.es (S.A.-S.); 2TECe19-Clinical and Translational Dermatology Investigation Group, Ibs.Granada, 18012 Granada, Spain; 3Cell Production and Tissue Engineering Unit, Virgen de las Nieves University Hospital, Andalusian Network of Design and Translation of Advanced Therapies, 18012 Granada, Spain; alvarosisan@gmail.com; 4Biosanitary Institute of Granada (ibs.GRANADA), 18012 Granada, Spain; 5Dermatology Department, Granada School of Medicine, Granada University, 18016 Granada, Spain

**Keywords:** androgenic alopecia, alopecia areata, mesenchymal stem cells, advanced medical therapies, tissue engineering, gene therapy

## Abstract

Alopecia is a challenging condition for both physicians and patients. Several topical, intralesional, oral, and surgical treatments have been developed in recent decades, but some of those therapies only provide partial improvement. Advanced medical therapies are medical products based on genes, cells, and/or tissue engineering products that have properties in regenerating, repairing, or replacing human tissue. In recent years, numerous applications have been described for advanced medical therapies. With this background, those therapies may have a role in the treatment of various types of alopecia such as alopecia areata and androgenic alopecia. The aim of this review is to provide dermatologists an overview of the different advanced medical therapies that have been applied in the treatment of alopecia, by reviewing clinical and basic research studies as well as ongoing clinical trials.

## 1. Introduction

Alopecia is one of the most common consultation requests in dermatological daily practice [1]. This condition is usually associated with some psychological disturbances such as anxiety, depression, and distress [2]. In recent decades, many topical, intralesional, oral, or surgical treatments have been employed in order to delay and stop the hair loss as well as to restore the presence of hair in alopecic areas. However, some of the therapeutic drugs employed in some conditions such as minoxidil, finasteride, and dutasteride for androgenic alopecia (AHA) only provide partial and temporary involvement. Moreover, treatments for severe alopecia areata (AA) and many forms of scarring alopecia are usually ineffective and sometimes are associated with serious adverse effects [3,4]. With this background, some alternative therapeutic strategies are needed for this disease.

The use of genes, cells, and tissues as a new therapeutic resource is one of the characteristics of contemporary medicine. Advances in regenerative medicine has increased interest in applying stem cells to engineered tissue scaffolds to reconstitute damaged tissue and develop regenerative therapies for the skin [5,6]. This new therapeutic approach is based on gene therapy, cell therapy, tissue engineering, or the combination of any type of drug with medical devices (Figure 1). As defined by the European Commission, Advanced Therapy Medicinal Products (ATMPs) are new medical products based on genes (i.e., recombinant nucleic acids, gene therapy), cells (cell therapy), and/or tissue engineered products that contain or consist of engineered cells or tissues and are presented as having properties for, or are used in or administered to, human beings with a view of regenerating, repairing, or replacing human tissue [7,8]. According to the Annex I to Directive 2001/83/EC, a somatic cell therapy medicinal product means a biological medicinal product that presents two main characteristics:(a)Contains or consists of cells or tissues that have been subject to substantial manipulation so that biological characteristics, physiological functions, or structural properties relevant for the intended clinical use have been altered, or of cells or tissues that are not intended to be used for the same essential functions in the recipient and the donor;(b)Is presented as having properties for, or is used in or administered to, human beings with a view of treating, preventing, or diagnosing a disease through the pharmacological, immunological, or metabolic action of its cells or tissues.

Regarding point (a), the European regulation has established which manipulations should not be considered as substantial manipulations. In this way, cellular products obtained by centrifugation, irradiation, cell separation, concentration, or purification proceedings are not considered as ATMPs. Thus, common cellular treatments for some forms of alopecia like platelet-rich plasma could not be included into the ATMPs. Autologous fat injections for the treatment of alopecia are considered as an ATMP because they are used for a different purpose than originally intended, although their use is currently only approved in the setting of clinical trials. However, its illegal use has become extended in several cosmetic medicine centers, which entails serious legal implications [9].

ATMPs are complex products and risks may differ according to the type of product, nature/characteristics of the starting materials, and level of complexity of the manufacturing process. It is also acknowledged that the finished product may entail some degree of variability due to the use of biological materials and/or complex manipulation steps. In that way, ATMPs’ quality plays a major role in their safety and efficacy profile. Compliance with Good Manufacturing Practice (“GMP”) is an essential part of the pharmaceutical quality system and it is the responsibility of the ATMP manufacturer to ensure that appropriate measures are put in place to safeguard the quality of the product [10].

ATMPs are usually well tolerated, but some side effects have been described, especially with intravenous human mesenchymal stem cell (hMSC) therapy. MSC-based products express variable levels of a highly procoagulant tissue factor (TF/CD142) that could lead to venous thrombosis and thromboembolism. Thirabanjasak et al. reported a lupus nephritis patient who had angio-myeloproliferative lesions after direct injection of stem cells into the renal parenchyma [11]. Thus, it is advisable to weigh the risks and benefits of this treatment in patients with procoagulant diseases and thrombophilias. Low-grade fever has also been associated with hMSC infusion [12,13]. More data on the safety of stem cell application need to be collected.

ATMPs have numerous potential applications, especially in the field of regenerative medicine. With this background, the aim of this review is to outline the different forms of ATMPs and their application in the treatment of non-scarring alopecia forms such as alopecia areata and androgenic alopecia, by reviewing clinical and basic research studies as well as ongoing clinical trials.

## 2. Mesenchymal Stem Cell Therapy

HMSC-based therapies have been used in regenerative medicine in several medical areas such as orthopedics, neurology, cardiology, and dermatology. HMSCs originate from the mesoderm and share their origins with the skin. When these cells are implanted in a damaged tissue, they are likely to react better to paracrine factors; therefore, hMSCs are an ideal strategy for repairing and regenerating skin abnormalities (Figure 2) [5,13,14,15].

When these cells are administered to an area affected by different diseases, hMSCs can be differentiated into injured tissue components. In addition, these cells have shown an important immunomodulatory activity and are able to secrete various cytokines and growth factors such as Il-6, Il-7, Il-8, vascular endothelial growth factor (VEGF), basic fibroblast growth factor (bFGF), and epidermal growth factor (EGF) that promote tissue regeneration [16,17,18,19]. Although the main reserve of hMSCs is found in the bone marrow, adipose tissue has been identified as a source of these cells, displaying similar properties to those extracted from the bone marrow [20,21,22].

HMSCs have been used in several medical and surgical disciplines in order to explore new therapeutic possibilities for diseases which current treatment modalities do not offer satisfactory results. For example, in the orthopedic field, hMSC have been successfully used for osteonecrosis, osteoarthritis, and to promote fracture healing [23,24,25,26]. In dermatology, the main use of hMSCs has been focused on the treatment of wounds and skin ulcers. In that way, preclinical studies have shown that the application of hMSCs accelerates the re-epithelialization of skin wounds. This is due to the promoting action of dermal fibroblast proliferation by direct cellular contact and by transforming growth factor beta (TGFβ) and bFGF secretion. In addition, its ability to differentiate into adipocytes provides a supportive architecture for dermal regeneration and re-epithelialization [27,28,29,30,31].

The hair follicle (HF) is a regenerating system, which physiologically undergoes cycles of growth (anagen), regression (catagen), and rest (telogen). HF has a niche for mature stem cells in the attachment region of arrector pili muscles, which contain epithelial and melanocyte stem cells. Another type of stem cells within the hair follicle is dermal papilla cells, probably originating from dermal condensation, which is the initial stage of hair follicle development [32,33]. During adult HF cycling, the signal between epithelial keratinocytes and underlying specialized hair follicle and dermal papilla mesenchymal cells induces stem cell proliferation and initiates the cascade of cell differentiation into the hair follicle cell lineages [34]. These cells also take part in the regeneration of the sebaceous glands.

Hair loss is determined by several factors such as hereditary conditions, hormonal disorders, autoimmunity diseases, nutritional deficiency, bacterial and fungal overgrowth, psychological factors, environmental elements, and aging. Some of these harming factors influence the hair cycle and reduce stem cell activity and HF recovery capacity [33]. Immunologic disturbances use HF as one of their main targets. When the HF immune privilege is collapsed, CD4 and CD8 T cells and natural killer (NK) cells accumulate around the autoantigens of the hair bulb and contribute to the development of AA [35,36]. Multiple models have shown the association between the development of several types of alopecia with the disruption of certain cytokines and proteins. Th1 cytokines and chemokines, such as interferon-gamma (IFN-γ), CXCL9, and CXCL10, are predominantly detected in AA lesions and might induce the collapse of the hair follicle immune privilege [36]. CK15 immunoreactivity, which has been described as a marker of telogen, is decreased in people with active AA, whereas it is present in AHA. Hair follicles in the frontal parts of the scalp exhibit a deficit of CD34 in AHA, and its expression is preserved in hair follicles of the occipital region. Interfollicular injection of autologous CD34+ cell-containing PRP preparation has shown a positive therapeutic effect in AHA patients [37,38,39]. Moreover, CD200, another marker of matrix cells, is poorly expressed in patchy alopecia, which may be a sign of the disappearance of the immune privilege and can contribute to pathogenesis [40,41]. The Wnt pathway and Wnt/beta-catenin signaling are known to increase mammalian hair growth. In that way, Leirós et al. demonstrated that androgens deregulated dermal papilla cell-secreted factors involved in normal HF stem cell differentiation via the inhibition of the Wnt pathway [42].

Some preclinical studies have shown promising results by employing hMSCs in the treatment of AA and AHA (Table 1). Byun et al. developed a pilot study in order to demonstrate the immunomodulatory effect of MSC in AA. The investigators employed intravenous MSC in AA-induced C3H/HeJ mice on days 1 and 7, and after 15 weeks of follow-up, 23% of mice in the MSC-treated group showed AA incidence, whereas extensive hair loss was observed in 91% mice in the control group. Serum samples also showed decreased IFNγ, CXCL9, and CXCL10 concentrations in the treated group. Moreover, histological analysis demonstrated less inflammatory cellular infiltration around the dermal papilla [35]. Later, Kim et al. launched another in vitro study with the aim of assessing the effect of hMSCs on the viability and proliferation of human dermal papilla cells (hDPC) via the activation of the JAK/STAT and Wnt/beta-catenin signaling pathways in an AA-model. The investigators employed bone marrow-derived hMSCs that were co-cultured with hDPC, showing a 120% increased hDPC proliferation compared to hDPC cultured alone. In addition, hMSC treatment augmented beta-catenin levels and induced much higher phosphorylation of STAT1 and STAT3, proteins that are associated with a prolonged anagen phase. Finally, the investigators observed an immunomodulatory effect in the treatment with hMSC by restoring a HF privilege that induced re-entry signaling from the telogen to anagen phase [43]. Recently, Bak et al. conducted a mice-based study in order to evaluate the effect of umbilical cord blood hMSCs on reacquisition of hDPC conduction ability to induce hair growth. After 6 weeks of hMSC injection, a complete hair regrowth was observed. Moreover, histological analysis revealed that hMSC promoted hair follicle re-entry in the early and middle anagen phase prematurely compared with the minoxidil group. In addition, hMSC treatment up-regulated beta-catenin and AKT pathways and enhanced growth factors production [44].

Few clinical studies with advanced hMSC therapies have been conducted for several forms of alopecia, probably because of ethical considerations and the high cost of launching these studies (Table 2). Firstly, a group of Egyptian authors published clinical data on the use of hMSCs in the treatment of AA and AHA. In 2011, this group conducted a pilot study in eight AA patients in order to evaluate the effect of cultured and enhanced follicular stem cells (FSC) in the treatment of this condition. The FSC were extracted from a 4mm skin punch biopsy from the scalp and the FSC were injected once in the affected areas. At the end of the 6-month evaluation period, 5/8 patients (62.5%) achieved an excellent response, graded as an improvement of 50% or more. Two patients developed a good response (10–50% of improvement), whereas one patient showed poor response [45]. Later, Elmaadawy et al. developed a blinded randomized clinical trial where the purpose was to evaluate the safety and efficacy of the use of autologous bone marrow-derived mononuclear cells (including hMSCs) obtained without substantial manipulation compared to advanced FSC treatment for the management of resistant cases of AA and AHA. Patients were divided into four groups: groups 1 (10 resistant AA patients) and 3 (10 resistant AHA patients) received a single injection of mononuclear cells and groups 2 (10 AA patients) and 4 (10 AHA patients) received a single injection of FSC. After six months of clinical, dermoscopic, and histopathologic follow up, all the patients showed very good or excellent responses, especially in female patients, although 45% of AA patients suffered recurrence after one year of follow up. Moreover, no significant differences in dermoscopic or histopathologic analysis were found between the two studied treatments. Those data showed promising clinical results with advanced hMSC therapies for AA and AHA, but the frequent relapses after one injection revealed the need of multiple sessions, especially in AA patients [46]. In 2015, a Chinese group conducted a phase 1/phase 2, open-label clinical trial with human cord blood stem cells (CB-SC) in nine patients with established AA. All the patients received a single treatment of intravenous CB-SC combined with a Stem Cell Educator (Tianhe Stem Cell Biotechnologies^®^, Jinan, China). In order to carry out this Stem Cell Education Therapy, the patient’s blood was passed through a Blood Cell Separator MCS+ (Haemonetics^®^, Braintree, MA, USA) for 6 to 7 h to isolate mononuclear cells in accordance with the manufacturer’s recommended protocol. The collected mononuclear cells were transferred into the device for exposure to allogeneic CB-SCs, and other blood components were automatically returned to the patient. After 2 to 3 h in the device, CB-SC-treated mononuclear cells were returned to the patient’s circulation. The authors stated that all the patients tolerated the treatment well with a lack of serious adverse effects. Hair regrowth was noted after 4 weeks in patients with patchy and total AA. Moreover, 3/4 of the universal AA patients showed short vellus hairs on the scalp in week 12. After a two-year follow-up, two patients achieved complete hair regrowth, with no relapses. Only one patient with universal AA failed to show a response with this therapy [47].

Regarding autologous fat injection obtained by lipoaspiration, a 9-patient case series was published in 2017. In this study, the investigators employed a scalp injection of adipose cells obtained by a liposuction surgical technique enriched with a stromal vascular fraction (SVF) in patients with AHA. After 24 weeks of follow up, an increase in the number of hairs/cm^2^ measured by TrichoScan was noted [48]. Later, another group employed autologous adipose-derived stromal vascular cells (ADSVCs) in twenty patients with AA. In this paper, a significant increase in hair density and hair diameter was noted after a 6-month follow-up. Moreover, a significant decrease in the number of extracted hairs measured by a pull test was also found [49]. Finally, an anecdotical case report has shown the possible positive effect of autologous fat injections for scarring alopecia [50].

Recently, some authors have published their experience with Rigenera^®^ technology in patients with AHA. After extracting three 3-mm scalp samples, the tissues were disaggregated by employing Rigeneracons^®^ and, after that, the solution was rotated in the Rigenera machine allowing them to obtain the micrografts. This treatment was directly infiltrated into the scalp of the 100 included patients as a mesotherapy and, after a two-month follow-up, TrichoScan^®^ revealed an increase in hair density and total hair count [51].

Some other clinical trials with advanced hMSC and cellular therapy have been carried out in the last decade. In the beginning of the 2010s, a multicentric North American phase 2 study was conducted in order to evaluate and compare the efficacy of injections of ex vivo cultured and enhanced occipital autologous dermal and epidermal cells vs. dermal cells alone—two non-HMSC cellular therapies—in patients with hair loss. However, no further results of this study have been published yet [52]. Currently, a Lebanese group is recruiting patients for an open-label non-randomized clinical trial aiming to assess the effect of adipose-derived cultured hMSCs obtained by lipoaspiration and to compare their efficacy with non-cultured hMSCs. This group has experience with the employment of adipose-derived hMSCs in patients with AA, a therapy which has shown good clinical results [49,53]. Moreover, some other clinical trials are also evaluating the effectiveness and safety of autologous ex vivo expanded dermal and epidermal cultured cells in subjects with AA and AHA. Those trials are still in a pre-therapeutic phase and no results are available [54,55,56].

## 3. Gene Therapy in Alopecia

Gene therapy is a new approach to discover and treat many diseases, including alopecia. Gene-based therapeutics are broadly defined as using a vector to introduce nucleic acids into cells with the aim of altering gene expression to prevent, halt, or reverse a pathological process [57]. The skin was one of the first targets for experimental gene transfer, as it is a superficial organ, easy to manipulate and observe [58]. In gene therapy techniques, genetic material is usually transferred using modified vectors directly into a subject’s epidermal tissue (in vivo) or indirectly (ex vivo). When ex vivo techniques are used, cells are removed from the host, they are genetically manipulated, and then, reconstituted into the subject’s skin [59].

Gene therapy through RNA interference has been considered one of the most recent and revolutionary approaches used in individualized therapy. In fact, gene silencing and knockdown by topical siRNA has rapidly developed in recent years and its application in gene therapy has become an attractive alternative for drug development [60]. RNA interference by small interfering RNA (siRNA) is a technique to suppress the expression of certain genes with a high specificity [61].

The use of topical siRNA is limited because of its low permeability through the epidermis, its high molecular weight, its negative charge, and its susceptibility to degradation by endogenous enzymes [62,63,64]. For this reason, efforts in topical siRNA delivery are being focused on chemical methods to prepared carrier molecules able to mask siRNA-negative charges, compress the siRNA molecule to make it smaller, and protect siRNA from degradation as well as the use of physical methods [65].

Regarding chemical methods, currently, carriers of siRNA can be classified into two categories: viral and non-viral. Non-viral vectors are preferred because they have less toxicity, less immunogenicity, and easier preparation, but they present a low-efficiency transient gene expression [66,67]. Non-viral carriers typically involve complexing of siRNA with different compounds such as polymers, cationic lipids, cell penetration peptides nanocarriers, or others. Using nanocarriers has been shown to efficiently encapsulate siRNA, providing protection against degradation and greatly improving the efficiency of delivery [68].

Besides the advancement in nanocarriers systems, the release of the siRNA into the cellular cytoplasm (site of biological activity) remains low range [69]. As an alternative, different physical methods have been proposed to facilitate molecular permeability through non-endosomal cellular penetration, transferring the siRNA directly to the cellular cytosol facilitating the endosomal escape of the siRNA [70]. Iontophoresis, microneedle array devices, and ultrasound technique are the physical methods most employed for improving efficient delivery of either naked or loaded siRNA into the skin and promotion of gene silencing [60].

Nakumura et al. reported the effective controlled delivery of small interfering RNA using biodegradable cationized gelatin microspheres in an animal model of alopecia areata and demonstrated the specific inhibition of target gene expression, resulting in a restoration of hair shaft elongation (Table 3). They firstly proved the dominant role of Th1 cells in the alopecic areas, as the infiltrating CD4 T lymphocytes around the hair follicles of patients with alopecia areata were primarily CCR5-positive with few CCR4-positive cells. After that, they tried to reveal the effect of cytokine therapy in C3H/HeJ mice, a mouse model of alopecia areata, by applying recombinant Il-4 and neutralizing anti-interferon antibody local injections. They found that both effectively treated alopecia in in C3H/HeJ mice and demonstrated that intralesional injection of Il-4 suppressed CD8 T cell infiltrates around the hair follicles and repressed enhanced interferon mRNA expression in the affected alopecic skin. The siRNA used in this work was targeted to T-box21 (Tbox transcription factor), in order to inhibit the transcription factor responsible for the cytokine Th1 production. Therefore, Th1 T-box21 siRNAs conjugated to cationized gelatin showed mitigating effects on alopecia in C3H/HeJ mice, resulting in the restoration of hair shaft elongation. In that way, they demonstrated the use of gelatin–small interfering RNA conjugates as an efficient and safe tool for alopecia areata [71].

Four microRNAs—miR-221, miR-125b, miR-106a, and miR-410—were proved to be upregulated in balding papilla cells. In that way, they could participate in the pathogenesis of male pattern baldness. Therefore, it was suggested these microRNAs were possible candidates for a gene therapy regarding the strong therapeutic potential of microRNAs and the easy accessibility of hair follicles for gene therapy [72].

## 4. Tissue Engineering in Alopecia

Tissue engineering is an interdisciplinary field combining scaffolds, cells, and biomolecular signals to treat skin lesions. Its main challenge is the reconstitution of fully organized and functional organ systems from dissociated cells that have been propagated under a defined tissue culture condition. This strategy may contribute to the treatment of deep skin injuries and to the understanding of skin regeneration. Many dermal–epidermal composites or skin equivalents have been described to use in the clinic but the inability of these skin constructs to regenerate epidermal appendages, such as hair follicles, sebaceous, and sweat glands remains a major challenge [73,74,75].

Lee et al. developed a simplified procedure to reconstitute hair-producing skin (Table 4). They obtained epidermal and dermal cells from newborn mice, and mixed them in different ratios. A high-density cellular suspension was prepared in drops of minimal volume on tissue culture inserts or wells. They allowed the cells to settle until a gel consistency was obtained and they seeded the cells on the collagen beside a porous matrix of crosslinked bovine tendon collagen and glycosaminoglycan and a silicone layer. Both constructs were grafted in full thickness skin wounds generated on the back of athymic mice. Hair germ started to appear eight days after grafting, progressed to hair peg, and complete hair follicles were observed four days later. The skin was in good condition after a twelve-month follow-up [76]. Furthermore, two reports showed hair follicle neogenesis using hair germs obtained by the previously described “organ germ method” [77]. Asakawa et al. generated bioengineered hair germs mixing epithelial and mesenchymal cells derived from mouse embryos within a collagen gel drop, generating a structure (called “hair germ”) with two cellular layers separated by a translucent region (Table 5). They observed the presence of mature hair follicles when these hair germs were transplanted ectopically into subrenal capsules. These bioengineered hair follicles had a normal histological structure with outer root sheath, inner root sheath, dermal papilla, hair matrix, and sebaceous glands and they were connected to the arrector pili muscle and nerve fibers and were able to produce hair shafts [78]. Toyoshima et al. also demonstrated fully functional orthotopic hair regeneration via intracutaneous transplantation of bioengineered hair follicle germs generated by epithelial and mesenchymal cells derived from mouse embryos [79]. Later, Qiao et al. showed a more developed morphological structure, called “proto hairs”. It presented hair-like characteristics as an inner mass of cells similar to the dermal papilla structure, surrounded by matrix-like keratinocytes and a partially keratinized substance that could produce a hair shaft [80].

After that, Osada et al. artificially prepared dermal papilla cells spheres by aggregation of mouse vibrissae follicles in a round-bottom 96-well low-binding plate (Table 6). These spheres expressed higher amounts of versican, an anagen dermal papilla marker, than dermal papilla cells in monolayer cultures and they induced hair follicle neogenesis for at least twenty-six passages. However, these methods obtain variable sizes of microtissues and cell number content. To resolve this problem, substratum materials to enhance dermal papillae cells aggregation, such as polyethylene-co-vinyl alcohol or polyvinyl alcohol membranes, were employed, inducing hair follicular neogenesis [87,88,89].

Moreover, Chermnykh et al. showed that human keratinocytes from outer root sheath and dermal papillae cells cultured in a 3D network of extracellular matrix proteins and collagen formed tubule-like structures in this skin-equivalent in vitro [81], but did not form complete hair follicles. Similarly, Sriwiriyanont et al. observed neofollicle formation in nude mice grafted with engineered skin substitutes containing murine dermal papillae cells and human keratinocytes in a collagen–glycosaminoglycan matrix, but not in those containing human dermal papillae cells and human keratinocytes [83].

Ehama et al. showed the formation of hair follicle-like structures using human primary cultures of foreskin- or adult-derived epidermal cells co-grafted with murine dermal papillae cells by a chamber assay. The innermost regions were similar to the hair cortex and medulla of mature human follicles but they did not show a bulge region, all the follicular epithelial layers, and versican was not expressed, suggesting that the differentiation process was altered [84]. At the same time, Kang et al. obtained spheroid microtissues by culturing human dermal papillae cells in a 96-well low-binding plate, and implanted them intradermically into nude mice using the “patch assay”, generating new hair follicles. However, these hair follicles were not observed when monolayer dermal papillae cell cultures were used, concluding that cultured human dermal papillae cells do not induce hair neogenesis unless changes in the culture conditions were made [85]. Later, Higgins et al. found that human hair follicle dermal cells can be interchanged with interfollicular fibroblasts and used as an alternative cell source for establishing the dermal component of engineered skin, both in vitro and in vivo. These authors established some in vitro skin constructs by incorporating into the collagenous dermal compartment: primary interfollicular dermal fibroblasts, hair follicle dermal papilla cells, or hair follicle dermal sheath cells. In vivo skins were established by mixing dermal cells and keratinocytes in chambers on top of immunologically compromised mice. They found that all fibroblast subtypes were capable of supporting the growth of overlying epithelial cells, both in vitro and in vivo, being hair follicle dermal sheath cells superior to fibroblasts in their capacity to influence the establishment of a basal lamina. They also evaluated the human dermal papilla cells’ transcriptome, observing that monolayer dermal papillae cells cultures showed the most important changes immediately after early outgrowths from dermal papillae explants, which suggests that human dermal papillae cells spheroids were able to initiate hair follicle morphogenesis, but the production of a complete hair follicle required additional signals [90,91]. Others investigators produced dermal papillae cell spheroids from cultured dermal papillae cells on a Matrigel™ (Corning Life Sciences, Corning, NY, USA) scaffold, in order to restore dermal papillae signature gene expression of NCAM, versican, and α-smooth muscle actin, markers that are lost during the monolayer culture. They also observed that these dermal papillae cell-spheres, combined with hair germinal matrix cells onto Matrigel-coated plates, produced colorless fiber-like structures in vitro [82].

Other groups have cultured dermal mesenchymal cells in 3D conditions as the organotypic method of culturing cells inside a scaffold. This gel was seeded with keratinocytes from interfollicular skin, superior outer root sheath, or inferior outer root sheath obtaining an in vitro bi-layered skin. Nevertheless, only the constructs containing superior outer root sheath keratinocytes showed hair follicle-like structures [92]. Thangapazham et al. used a dermal equivalent composed of dermal papillae cells from human scalp contained in a collagen-I gel. Eight weeks after grafting onto nude mice, these constructs presented hair follicles showing bulb, dermal sheath, hair matrix, and cortex. Histological analysis showed concentric layers of inner root sheath and outer root sheath, sebaceous glands, and hair shaft. Immunohistochemistry assays revealed that both epithelial and dermal cells from neofollicles were of human origin, and dermal papillae cells and dermal sheath cells expressed human nestin and versican [86]. Leirós et al. showed that both epithelial and dermal cultured cells from adult human scalp in a dermal scaffold were able to produce in vivo structures that recapitulate embryonic hair development. In fourteen days, histological structures reminiscent of many different stages of embryonic hair follicle development were observed in the grafted area. These structures showed concentric cellular layers of human origin, and expressed K6hf, a keratin present in epithelial cells of the companion layer. However, the presence of fully mature hair follicles was not observed [93].

Recently, Abaci et al. regenerated for the first time a truly functional human skin in an entirely ex vivo context that incorporated hair follicles from cultured human cells. They used a biomimetic approach for generation of human hair follicles within human skin constructs by recapitulating the physiological 3D organization of cells in the hair follicle microenvironment using 3D-printed molds. Overexpression of Lef-1 in dermal papilla cells restored the intact dermal papilla cells’ transcriptional signature and enhanced the efficiency of hair follicle differentiation in human skin constructs. After that, vascularization of hair follicle-bearing human skin constructs increased graft survival and enabled efficient human hair growth in mice. They were able to generate 225 hair follicles, starting from only one hair follicles donor tissue, which was not possible with previous techniques [94].

Some clinical trials with tissue engineered therapies for alopecia have been developed in recent years. A Taiwanese group designed an observational study with 400 healthy adults that were going to receive cosmetic surgery such as removal of moles. The aim of this study was to note if a sample of in vitro cultured dermal papilla cells was able to induce hair follicle formation after maintaining their spherical structure before transplanting into dermis in vivo. To date, the authors have not published the results of this study [95].

## 5. Conclusions

Alopecia areata and androgenic alopecia are still challenging conditions for dermatologists, with a lack of highly effect treatments. With this background, advanced therapies are a promising therapeutic option that have shown good results in preclinical studies. However, more clinical studies are needed to verify if ATMPs can be a safe and effective treatment for diverse forms of non-scarring alopecia.

## Figures and Tables

**Figure 1 ijms-21-08390-f001:**
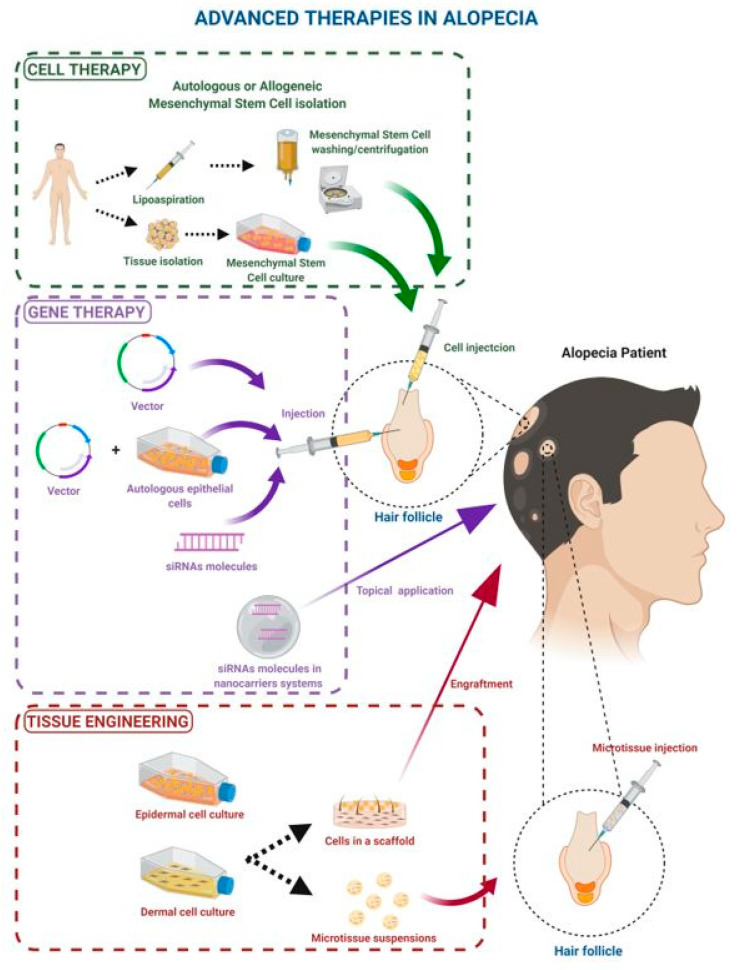
Advanced therapies and their use in alopecia: cell therapy, gene therapy, and tissue engineering therapy. Created with Biorender.com.

**Figure 2 ijms-21-08390-f002:**
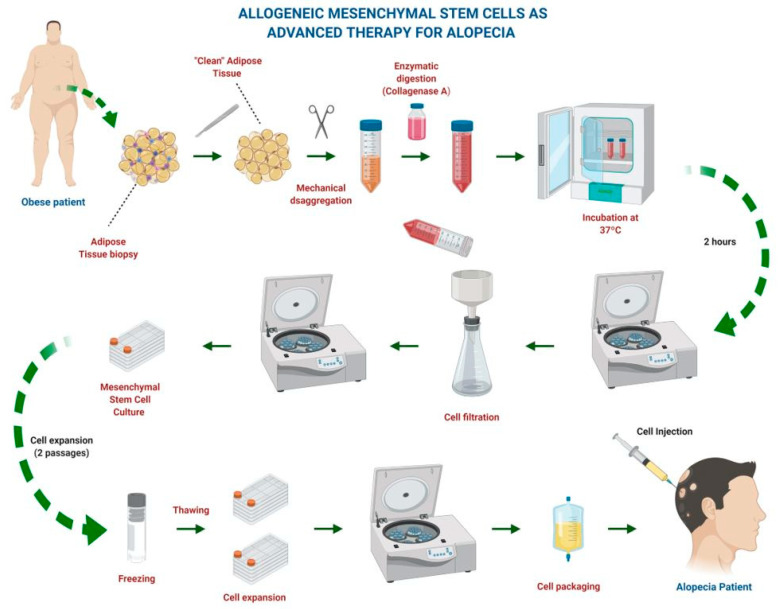
Procurement and production of mesenchymal stem cells. Created with Biorender.com.

**Table 1 ijms-21-08390-t001:** Preclinical studies of MSCs as advanced cell therapy for non-scarring alopecia.

MSC Source	Type of Study	Experimental Design	Results	Conclusions	Reference
Human bone marrow (hBM-MSCs)	In vivo study Alopecia Areata induced in 24 C3H/HeJ mice.	Control group (*n* = 11) and MSC group (*n* = 13). Intravenous injection of two doses of MSCs at days 1 and 7. Mice were analysed for 15 weeks.	1-MSC-treated mice showed decreased IFNG concentration as early as 5 weeks post-transplantation. 2-Mice treated with MSCs showed less inflammatory cell infiltration in the dermis and the number of hair follicles was not decreased compared with the control group.	1-After 15 weeks post-transplantation; 91% of control mice experimented hair loss against 23% of MSC-treated group. 2-MSCs mediate inhibition of IFNG and CD3 and CD8 + NKG2D+ T-cell infiltration which protects against the collapse of the hair follicle immune privilege.	[35]
Human bone marrow (hBM-MSCs)	In vitro study in Alopecia Areata induced environment.	Co-culture of human dermal papilla cells (hDPCs) (1 × 10^5^ cells per well) pre-treated with interferon gamma (IFN-γ) to reproduce Alopecia Areata environment with hBM-MSCs (5 × 10^4^ cells).	1-hDPCs proliferation were increased up to ~120% compared to control cultures. 2-Levels of β-catenin protein which were suppressed by IFN-γ treatment were shown to be reversed by hBM-MSCs. 3-hMSCs induced higher phosphorylation of STAT1 and STAT3. 4-hMSCs increased mRNA expression of TNF-α and IL-1β 5-Growth factor analysis revealed that co-culture of hBM-MSCs and hDPCs enhanced expression of IGF-2, TGF-α, TGF-β3 and NT-4 which could be involved in driving anagen reentry.	1-hHMSC co-culture could reverse suppressed hDPC proliferation and may prolong the anagen phase through activation of the β-catenin/Wnt and JAK/STAT signalling pathways. 2-hBM-MSCs in refractory AA patients might restore hair follicles (HFs) immune privilege through immunomodulatory effects and induce reentry signalling from the telogen to anagen phase	[43]
Human umbilical cord blood (hUCB-MSCs)	In vivo study in C3H/HeJ mice (Telogen-anagen transition model was induced by depilating the dorsal skin of mice in the telogen phase of the hair cycle).	Four groups of study (6 weeks): Control group Minoxidil-treated group Saline intra-dermal injection-treated group hUCB-MSC intra-dermal injection-treated group (8 sites—1 × 10^5^ cells/mice)	1-After 6 weeks post-treatment, hair regrowth was complete in the hUCB-MSCs group. 2-Minoxidil-treated group presented incomplete pigmentation and contained hairs in the early stage of the hair cycle, 3-Control and Saline-injected groups retained large areas without anagen induction.	1-hUCB-MSCs can accelerate the initiation of the hair follicle telogen-anagen transition, increase the number of hairs in vivo, and enhance expression of proteins related to hair induction in vitro. 2-IGFBP-1 (assumed as the main secretory factor of hUCB-MSCs) restores and promotes the hair-induction ability of hDPCs via an IGF-1/IGFBP-1 co-localization.	[44]
	In vitro analysis of the effect hUCB-MSCs co-cultured with human dermal papilla cells (hDPCs).	Co-culture of 1 × 10^5^ hDPCs with 2 × 10^5^ hUCB-MSCs.	1-hUCB-MSCs co-culture enhanced hDPC proliferation and restored ALP activity after 5 days, both of which are anagen markers of the hair cycle. 2-β-catenin, AKT, and GSK3β, which are proteins contained in the pathway related to cell growth and proliferation, were up-regulated in hDPCs by co-culture with hUCB-MSCs. 3-IGFBP-1 and VEGF were upregulated in the medium of the hDPC plus hUCB-MSCs group compared to the hDPC group.		

**Table 2 ijms-21-08390-t002:** Clinical studies of MSCs as advanced cell therapy for non-scarring alopecia.

MSC Source/Clinical Trial Title	Experimental Design	Pre-Treatment Data	Clinical Results	Conclusions	Reference
Autologous hair follicle stem cells from the lower bulge areas	N = 8 patients with Alopecia Areata (AA). One millilitre (in a density of 10^5^ cells/mL) was injected intradermally once per centimetre square using a 23-gauge needle. Clinical improvement was assessed by calculating the percentage of the difference in the AA at the end of 3 months and 6 months in relation to the baseline extent.	The extent of affection ranged from 25 to 70% (mean: 48 ± 15%). Patients age ranged from 6 to 17 years. (mean: 10.12 ± 3.68 years). The duration of disease ranged from 1 to 4 years (mean: 2.31 ± 0.96 years)	1-Patients showed variable degrees of response, 20–80% (mean: 45 ± 22%) from baseline at the end of the third month and 30–100% (mean: 69 ± 27%) from baseline at the end of the sixth month. 2-After 6 months, excellent response was achieved in five patients (62.5%), good response was achieved in two patients (25%), whereas one patient (12.5%) showed poor response. 3-There was a negative correlation between the age of the patients and the grade of response they achieved, that is, the younger the patient, the better the response.	1-Approximately 60% of the patients (the excellent responders) reported improved quality of life.	[45]
Autologous bone marrow derived mononuclear cells (including stem cells) and follicular stems cells	N = 40 patients: 20 patients with resistant Alopecia Areata (AA) and 20 patients with Androgenetic Alopecia (AGA). Four groups of study: *Group 1 (10 resistant AA patients) *Group 3 (10 AGA patients) *These received a single session of intradermal injections of autologous bone marrow derived mononuclear cells (BMMCs). **Group 2 (10 resistant AA patients) **Group 4 (10 AGA patients) **These received a single session of intradermal injections of autologous follicular stem cells (FSC). One millilitre (in a density of 10^5^ cells/mL) was injected intradermally once per centimetre square using a 26-gauge needle. Clinical improvement was assessed at 3 and 6 months	The extent of affection ranged from; Mean Group 1: 44 ± 20.65% Mean Group 2: 46± 18.97% Mean Group 3: 58± 19.32% Mean Group 4: 50 ± 0% The ages of the patients ranged from; 10-50 years (mean 26 ± 8 years). All patients were resistant to conventional treatments. The duration of disease ranged from; Mean Group 1: 2 ± 1 years Mean Group 2: 1.5 ± 0.7 years Mean Group 3: 2 ± 0.9 years Mean Group 4: 1.9 ± 0.8 years	1-The mean percentage of improvement in AA patients was 45 ± 22% in AA subjects receiving autologous BMMCs and 58 ± 34% in AA subjects receiving autologous FSC with non-statistically significant difference. 2-The mean percentage of improvement in AGA patients was 52 ± 28% in AGA subjects receiving autologous BMMCs and 42 ± 27% in AGA subjects receiving autologous FSCs with non-statistically significant difference. 3-45% of AA patients suffered recurrence of disease activity after one year of follow up mostly due to their stressful life events.	1-BMMCs and FSCs gave significant improvement in AGA and AA with no statistically significant difference between both methods. 2-Autologous BMMCs and autologous FSC seem to be a safe, tolerable and effective treatment	[46]
Autologous blood mononuclear cells “educated” with allogeneic umbilical cord blood stem cells Stem Cell Educator Therapy	N = 9 patients:9 alopecia areata patients (AA). All of them received a Stem Cell Educator Therapy consisting in autologous blood mononuclear cells isolated using a Blood Cell Separator MCS+ (6–7 h) and then, transferred into the Educator device for exposure to allogeneic cord blood stem cells (CB-SCs) ex vivo cultured and prepared (2–3 h). After this time, CB-SC-treated mononuclear cells were returned to the patient’s circulation via a dorsal vein in the hand (16-gauge IV needle) with physiological saline (2 to 3 mL/min). Follow-up visits were scheduled 4, 12, 24, 40, 56, 84, and 112 weeks after treatment for clinical assessments and laboratory tests. Skin biopsies of scalps were performed before the treatment and at 12 weeks post-treatment.	The degree of affection ranged from patchy alopecia areata (3), alopecia totalis (2) to alopecia universalis (4). Mean alopecic duration: 5.1 ± 6.1 years. The ages of the patients ranged from; 12–26 years (mean 20.1 ± 4.6 years).	1-At 4 weeks post-treatment with Stem Cell Educator therapy, there was hair regrowth in subjects with patchy AA and alopecia totalis. 2-Patients (3/4) with alopecia universalis exhibited regrowth of eyebrows and eyelashes at the 12-week follow-up. 3-All these improvements were maintained throughout the final follow-up at 2 years. 5-Only one participant with alopecia universalis failed to show a response to the Stem Cell Educator therapy, possibly due to a previous long-term therapy with oral prednisone	1-This phase 1/phase 2 study demonstrates the safety and feasibility of Stem Cell Educator therapy in the treatment of AA subjects. 2-Stem Cell Educator therapy can control the autoimmunity and lead to hair regrowth.	[47]
Autologous adipose cell enriched with stromal vascular fraction (SVF)	N = 9 patientswith androgenic alopecia (AHA). The adipose cells were injected into the scalp of the patient in a single dose. 1 mL/cm^2^ scalp was injected. The composition was a mixture of adipose-enriched SVF cells and Lactated Ringer. Follow-up for hair count, anagen percentage, telogen percentage and cumulative thickness at weeks 6, 12 and 24.	Degree of affection: Hamilton II-VI/Ludwig I-III Average age: 29 (range 19–54) Average volume of adipose cells injected: 30 mL	1-Hair count was significantly augmented after 24 weeks (28.4 ± 17.3; *p* = 0.010). 2-Percentage of anagen hair was increased but did not reach a statistical significance (*p* = 0.094). 3-Percentage of telogen hair was decreased but did not reach a statistical significance (*p* = 0.094). 4-Cumulative thickness was augmented but did not reach a statistical significance (*p* = 0.133).	1-Enriched adipose cells injections may be a promising approach for treating AHA in both men and women.	[48]
Autologous adipose-derived stromal vascular cells (ADSVCs)	N = 20 patients with confirmed diagnostic of hair loss. The ADSVCs were injected into the scalp of the patient with a 30-gauge needle. A total of 5 mL was injected in 25 spots. 4 to 4.7 × 10^6^ cells were transplanted: in fact, 0.2 mL containing 0.160–0.188 × 10^6^ cells were injected per spot (total = 25 spots, 5 mL) Follow-up for hair evaluation was based on the hair cycles and was performed 1 week, 3 months, and 6 months after the procedure.	All the selected subjects showed partial alopecia grade 1 or 2 at Ludwig Scale. The ages of the patients ranged from; 20-63 years(mean 38.3 ± 2.3 years). 55% of the patients showed medium diameter hair and 45% showed fine hair. Study subjects showed abnormal hair density (density < 175 hair/ cm^2^ in 100% of the subjects)	1-Hair diameter increased significantly, especially 6 months after the treatment (80.8 ± 2.4 μ and 62.8 ± 1.7 μ vs. 60.5 ± 1.8 μ for 6 and 3 months postoperatively vs. preoperatively). 2-Hair density was significantly augmented after treatment (121.1 ± 12.5 and 120.8 ± 12.6 vs. 85.1 ± 8.7 for 6 and 3 months postoperatively vs. preoperatively). 3-Results of the pull test showed a significant decrease in the number of extracted hair (0.80 ± 0.17 and 0.90 ± 0.20 vs. 4.35 ± 0.33 for 6 and 3 months postoperatively vs. preoperatively). 4-2 of the 20 patients showed no significant improvements.	1-ADSVC injection promotes good stability of the hair by increasing the hair density, the hair diameter, and decreasing the pull test to almost zero.	[49,50,51,52]

**Table 3 ijms-21-08390-t003:** Preclinical studies of Gene Therapy for non-scarring alopecia.

Type of Gene Therapy	Type of Study	Experimental Design	Results	Conclusions	Reference
Intralesional injections of oligonucleotides and siRNAs	In vivo study Older female C3H/HeJ mice having an alopecic region on the back in the waxing phase.	N = 78 C3H/HeJ mice with alopecic lesions (develop hair loss spontaneously): Group 1: Treatment with Il4 injections (*n* = 10) Group 2: Treatment with 0.9% sodium chloride injections (*n* = 10) Group 3: Treatment with Il4 and Ifng injections (*n* = 10) Group 4: Treatment with anti-Ifng antibody injections (*n* = 6) Group 5: Treatment with rat IgG injections (*n* = 6) Group 6: Treatment with antisense *Tbx21* oligonucleotide injections (*n* = 6) Group 7: Treatment with non-sense oligonucleotide injections (*n* = 6) Group 8: Treatment with cationized gelatin-conjugated *Tbx21* siRNA injections (*n* = 8) Group 9: Treatment with naked *Tbx21* siRNA injections (*n* = 8) Group 10: Treatment with cationized gelatin-conjugated non-sense siRNA injections (*n* = 8)	1-Intralesional injections of recombinant Il4 (0.1 μg) (Group 1) every day for 3 weeks significantly restored hair shaft elongations when compared with the sodium chloride injections (Group 2). There was no recurrence of alopecia from these mice during a 2-month. 2-Il4 effect was suppressed by the simultaneous injection of 0.01 μg of recombinant Ifng (Group 3). 3-Anti-Ifng antibody injections (Group 4) improved the hair growth index more efficiently than the control rat IgG (Group 5). There was no disappearance of hair shafts from these mice during a 2-month observation period after the cessation of antibody application. 4-Antisense *Tbx21* oligonucleotide (Group 6) was significantly more effective for alopecia than non-sense oligonucleotide (Group 7). 5-Cationized gelatin-conjugated *Tbx21* siRNA injections (Group 8) were more effective than naked *Tbx21* siRNA injections (Group 9) or non-sense siRNA conjugated with cationized gelatin (Group 10). There was no recurrence of alopecia in the mice during a 2-month observation period after the cessation of *Tbx21* siRNA application.	1-Intralesional injections of Il4 suppressed an enhanced expression of *Ifng* in alopecic skin. 2-Intralesional injections of *Tbx21* antisense oligonucleotide restored the hair shaft elongation. 3-Efficient and safety delivery of *Tbx21* siRNA to alopecic skin using a biodegradable cationized gelatin demonstrated specific inhibition of target gene expression (*Ifng*) resulting in a restoration of hair shaft elongation.	[71]

**Table 4 ijms-21-08390-t004:** Preclinical in vitro studies of Tissue Engineering for non-scarring alopecia.

Cell Types and Scaffold	Type of Study	Experimental Design	Results	Conclusions	Reference
Human Dermal Papilla (DP) cells and Keratinocytes (KT) in type I collagen matrix	In vitro study	DP cells were isolated from hair follicle of human scalp. Rat vibrissa DP cells were isolated with microdissection. Keratinocytes were isolated from normal human skin. Dermal equivalents were prepared with type I collagen extracted from rat tail tendons. Human dermal fibroblasts or DP cells were embedded in a final concentration of 1 × 10^5^ cells/mL were added. Keratinocytes were seeded on the top of the gel or embedded inside the collagen matrix.	1-Human and rat vibrissa DP cells were able to reorganize the collagen lattices within the first 48 h. Contraction was significantly stronger with rat vibrissa DP cells than with human DP cells (60 and 40%, respectively). 2-Addition of epidermal keratinocytes enhanced contraction in both cases (75 and 56%, respectively). 3-After 1 week, rat vibrissa DP cells cultured together with keratinocytes totally disaggregated and lysed collagen lattices. 4-Human DP cells reorganized collagen matrix but were unable to disintegrate it. 5-After 10 days, DP cells embedded into collagen gel, and keratinocytes seeded on the top of the gel formed tubular structures.	1-DP cells induced formation of multicellular tube-like outgrowths in the culture of epidermal keratinocytes	[81]
Human dermal papilla (DP) cells cultured in Matrigel	In vitro study	Dermal cells were isolated from dermal papillae microdissected from the bulbs of dissected hair follicles. Cells were seeded onto wells precoated with Matrigel or hyaluronan or not. For the formation of DP spheroids in 3D Matrigel culture, different conditions were evaluated. All experiments, characterization of different culture conditions was conducted after 5 days of harvesting.	1-When 1 × 10^4^ DP cells were cultured on the 96-well plates precoated with Matrigel for 5 days, both passage 2 and passage 8 DP cells formed spheroidal microtissues with a diameter of 150–250 μm. 2-Cells within DP spheres could disaggregate and migrate out, which was similar to primary DP. 3-Expression of several genes and proteins associated with hair follicle inductivity of DP cells, such as NCAM, Versican, and α-smooth muscle actin was elevated in the spheres compared with the dissociated DP cells. 4-DP spheroids mixed with HGMCs (hair germinal matrix cells—commercial cell line) and incubated on the Matrigel surface, developed colourless hair shafts.	1-3D Matrigel culture technique is an ideal culture model for forming DP spheroids and that sphere formation partially models the intact DP, resulting in hair induction, even by high-passage DP cells.	[82]

**Table 5 ijms-21-08390-t005:** Preclinical in vivo studies of Tissue Engineering for non-scarring alopecia.

Cell Types and Scaffold	Type of Study	Experimental Design	Results	Conclusions	Reference
Mouse epidermal and dermal cells in a gel-like endogenous matrix or in Integra™ Bilayer Wound Matrix	In vivo study Athymic nude, hairy SCID, or normal mice of the same inbred strain	Dissociated new-born mouse epidermal and dermal stem cell were mixed in different ratios. For gel-like endogenous matrix: 150–200 μL of cell suspension, containing 2–20 million cells, was pipetted onto a tissue cell culture insert. For each 1.5 cm^2^ piece of Integra, 12 million epidermal cells and 60 million dermal cells in 200 μL of serum-free medium. The intended area of skin to be grafted for hair bearing was excised in full thickness.	1-At day 8, hair germs started to appear, which progress to the hair peg stage at about day 9. 2-Hairs can be seen on the surface of the wound as early as 11–15 days postgraft. 3-Histological sections of the skin at day 11 postgraft showed that normal layers of the skin were regenerated. 4-after hairs were clipped or plucked, they grew and reached normal length in about 2 months.	1-Ratio of 1:5–10 for epidermal:dermal cells is optimal. 2-There are no differences of skin quality or hair growth when tissue culture inserts or Integra^TM^ are used. 3-After 18 months, hair growth and cycling are active.	[76]
Mouse embryonic epithelial and dermal cells using the organ germ method. Collagen gel	In vivo study -C57BL/6 mice -C57BL/6-TgN (act-EGFP) mice -C57BL/6-TgN (act-EGFP) OsbC14-Y01-FM131 mice -Balb/c nu/nu mice	Bioengineered hair follicle germs: 7.5 × 10^4^ epithelial cells and 7.5 × 10^4^ mesenchymal cells which were derived from skin from C57BL/6-TgN (act-EGFP) mouse embryos. After 2 days in culture, to develop and mature bioengineered hair follicles, they were transplanted into the sub-renal capsules of 8-week-old C57BL/6 mice At 14 days after engraftment, mature bioengineered hair follicles were harvested and dissected into a single or a couple of follicular units via stereomicroscopic observation. Mature bioengineered hair follicles were intracutaneously grafted into Balb/c nu/nu mice.	1-After 7 to 10 days after the orthotopic transplantation, the wound was completely healed. 2-The eruption of the bioengineered pelage shaft was observed at 14 ± 1.8 (*n* = 30) days at a frequency of 90% (*n* = 33) 3-The bioengineered hairs repeatedly exhibited growth and regression. 4-The periods of growth and regression of bioengineered hair follicles lasted 11.0 (±2.6) days and 9.4 (±2.4) days, respectively.	1-Bioengineered hair follicles generated by ectopic transplantation can functionally replace orthotopic FUT therapy. 2-Ectopic bioengineered pelage follicle connected to the epidermal layer of the skin, reproduced the stem cell niche and the hair cycle equivalent to the natural pelage, and repeatedly produced the same hair types during the hair cycles.	[78]
Mouse embryonic skin epithelial and mesenchymal cells. Epithelial cells from adult vibrissa-derived bulge region and primary cultured dermal papilla (DP) cells. Collagen gel	In vivo study -C57BL/6 -C57BL/6-TgN (act-EGFP) OsbC14-Y01-FM131 mice -Balb/c nu/nu mice	The bioengineered pelage follicle germs: mouse embryonic skin epithelial and mesenchymal cells (7.5 × 10^3^ of each cell type) The bioengineered vibrissae follicle germ: epithelial cells (1 × 10^4^ cells) isolated from the adult vibrissae-derived bulge region and primary cultured DP cells (3 × 10^3^ cells) Bioengineered hair germs were ectopically engrafted into the sub-renal capsules of C57BL/6 mice or intracutaneously transplanted onto the back of Balb/c mice.	1-Eruption and growth hair shafts were observed at a frequency of 94% (*n* = 78) and 74% (*n* = 62) for bioengineered pelage and vibrissae follicles, respectively. 2-Bioengineered pelage follicle and the vibrissae follicle formed correct structures comprising an infundibulum and sebaceous gland in the proximal region. 3-Bioengineered pelage follicle germs were found to produce all types of pelage hairs. 4-The bioengineered pelage and vibrissae follicles repeated the hair cycle at least 3 times during the 80-day period.	1-Both bioengineered hair follicles produce follicles that can repeat the hair cycle, connect properly with surrounding skin tissues and achieve piloerection.	[79]
Human Dermal Papilla (DP) cells and Keratinocytes (KT) in collagen-glycosaminoglycan scaffolds	In vivo study	Human or murine DP cells, combined with foreskin-derived keratinocytes (KT) or transduced KT with pBABE-puro encoding N-terminally truncated β-catenin (KT’) (expression induced by hydroxytamoxifen (4OHT) administration) A positive control fabricated with murine hair from newborn cells was also evaluated. After 10 days incubation at air–liquid interface, Engineering Skin Substitutes (ESS) were grafted to athymic mice and were evaluated for 6 weeks.	1-*EF1* and *WNT10B* were significantly higher in 4OHT-treated ESS compared with vehicle-treated ESS but no hairs were observed in ESS with KTs’ and hDP cells. 2-Only ESS with mDP cells formed follicular structures, as confirmed by trichohyalin and keratin 10 immunostaining.	1-Chimeric hair follicles were successfully generated in ESS containing combinations of mDP cells and KTs or KTs’, although they were deficient anatomically. 2-DP cells play an important role in the induction of hair morphogenesis in ESS.	[83]
Mouse Dermal Papilla (DP) cells and human keratinocytes in grafting chambers	In vivo study -Versican-GFP transgenic mice -Nude mice (bulb/c, nu/nu)	DP cells were isolated from skin of Versican-GPF transgenic mice. Keratinocytes were isolated from human scalps tissues and neonatal foreskins. Human epidermal cells and DP cell fractions (containing 1 × 10^6^–10^7^ cells of each) were transferred to grafting chambers implanted on the dorsal skins of nude mice (bulb/c, nu/nu). Chambers were removed 1 week after grafting, and hair follicle formation was assessed at 3–4 weeks.	1-DP cells were able to induce hair follicles together with the epidermal component but only when both epithelial and mesenchymal components were present. Hair pegs were formed a week after grafting. 2-When the number of epidermal cells was reduced to 1 × 10^6^ cells (10% of DP cells), the efficiency of hair follicle reconstitution was mostly unchanged. 3-When human keratinocytes were included, hair follicle-like structures were formed at the graft sites 4 weeks later and innermost regions of the structures were clearly keratinized. 4-Human adult cells also have the same ability to differentiate into follicular keratinocytes as neonatal foreskin-derived epidermal cells.	1-Results show that hair follicle-like structures consisting of human keratinocytes and murine mesenchymal cells are generated. 2-Epithelial-mesenchymal interactions exists between human and mouse cells.	[84]
Mouse dermal and human epidermal cells. Patch assay	In vivo study	Mouse dermal and epidermal cells were freshly isolated from C57BL/6 used for control experiments. Human dermal papilla (DP) spheres (10^4^ cells) were prepared from two-dimensional (2D) cultured DP cells using either low cell-binding plate or hydrocell plate and combined with freshly isolated mouse epidermal cells for implantation A total of 200 DP spheres (2 × 10^6^ cells) prepared from human DP cells were mixed with fresh mouse epidermal cells (2 × 10^6^ cells) and implanted. 50 DP spheres (5 × 10^5^ cells) prepared from human DP cells were mixed with fresh mouse epidermal cells (5 × 10^5^ cells) and implanted. Mice were killed 2 weeks after cell implantation in order to verify hair follicle induction.	1-Hair follicle was observed in positive control experiments with mouse dermal and epidermal cells. 2-Hair follicle formation was observed when human DP spheres from various passages of culture were mixed with new born mouse epidermal cells. 3-Hair follicles were never observed when 2D cultures from the same population were use 4-The morphology and size of hair follicles induced by human DP spheres resembled the ones induced by mouse dermal cells.	1-Using a reconstitution assay, sphere formation increases the ability of cultured human DP cells to induce hair follicles from mouse epidermal cells	[85]
Human Dermal Papilla (DP) cells embedded into rat tail collagen type 1 and neonatal foreskin keratinocytes (NFK) Dermal-epidermal composites (DEC)	In vivo study -Nude mice	Human DP cells were isolated from temporal scalp dermis. DECs were constructed by combining DP cells with rat tail collagen type 1, adding NFKs on top and bringing the constructs to the air-liquid interface for 2 days before grafting onto female nude mice.	1-Alkaline phosphatase activity was variable between samples, with cells from 3 of the donors showing alkaline phosphatase activity in more than 50% of the cells. 2-8 weeks after grafting, hair follicles (HFs) were observed in mice grafted with the 3 human DP cells with higher alkaline phosphatase activity. 3-HFs had a bulb, dermal sheath, hair matrix and cortex 4-Cells in the region of the DP and displayed alkaline phosphatase activity, normal reactivity with specific antibodies to human nestin and versican. 5-Basal layer of the outer root sheath was immunoreactive for keratin 15.	1-Cultured specialized human cells such as DP cells can induce complete pilosebaceous units in vivo in the grafted DEC model.	[86]

**Table 6 ijms-21-08390-t006:** Preclinical in vitro and in vivo studies of Tissue Engineering for non-scarring alopecia.

Cell Types and Scaffold	Type of Study	Experimental Design	Results	Conclusions	Reference
Mouse embryonic skin epidermal and dermal cells. Cell aggregates	In vitro and in vivo study -C57BL/6 mice -Male CD-1 nude albino mice	Mixed dermal and epidermal cells (keratinocytes and melanocytes) were removed from embryonic day 18 C57BL/6 mice. Aggregates were formed using the hanging droplet method. 2 × 10^6^ cells/mL were hanging droplet in 20 µL. Aggregate formation was completed within 18–20 h. To form proto-hairs, aggregates were transferred individually to wells of a 96-well round-bottom plate. Wells were precoated with 0.24% methylcellulose medium to prevent adherence of proto-hairs. A single cultured aggregate (proto-hair) was grafted into the ear of male CD-1 nude mice. Regrowth of neo-generated hairs was regularly monitored.	1-Few days after aggregate formation (4-7 days), hair-like structures started forming. 2-The frequency of hair morphogenesis ranged from 66 to 100% among numerous experiments. 3-Proto-hairs could undergo further maturation in vivo. 4-Within 2 weeks, black-pigmented hair fibres appeared. 5-Within 4 weeks after implantation, approximately 50–60% of implants developed follicles with pigmented hair shafts. 6-Hairs developed from implanted proto-hairs were well anchored in the skin and persisted for at least 6 months 7-Implanted proto-hairs were able to maintain their growth for many months.	1-In vitro incubation of mixed follicular cell aggregates leads to the formation of partially developed follicle-like structures called proto-hairs. 2-Upon implantation, proto-hairs fully develop into normal hairs that persist and grow indefinitely.	[80]
Mouse, rat and human Dermal Papilla (DP) cells. Spheres	In vitro and in vivo studies -Versican-GFP-transgenic (versican-GFP-Tg) mice -C57BL/6J -Wistar rats -Athymic nude mice	DP cells were dissected from mouse vibrissae follicles (versican-GFP-Tg) and 10^4^ DP cells were aggregated to form one spherical structure and maintained for 2 to 12 days. Epidermal and DP cells were dissected from C57BL/6J mouse. Fifty of the DP spheres or 5 × 10^5^ of the dissociated DP cells were combined with 5 × 10^5^ epidermal cells. Cells were injected subcutaneously into athymic nude mice. Mice were killed 2 weeks after cellimplantation to verify hair follicle induction	1-Hair follicle induction could occur even from DPspheres of cells after 26 passages 2-DP cell suspensions from more than 8 passages, however, could not induce hair follicles.	1-DP have highly aggregative properties compared withskin fibroblasts 2-DP spheres may have recovered their aggregative property by increasing their versican content, which helped them to interact with epidermal cells and induce hair follicles.	[87]
		DP cells were isolated by scissors and forceps from cheeks of Wistar rats- 10^4^, 2 × 10^4^, 4 × 10^4^, 8 × 10^4^ and 1.6 × 10^5^ cells were added to each well, containing poly-(ethylene-co-vinyl alcohol) (EVAL). DP microtissues were characterized in vitro and In vivo.	1-Formation of DP microtissues on EVAL is affected by seeding cell numbers. 2-After 5 days of culture, dense microtissues were observed on EVAL at the seeding numbers of 8 × 10^4^ cells/well or higher. 3-About 47 microtissues (diameter > 125 mm) were obtained on an EVAL surface of 1.9 cm^2^ with a single seeding of 1.6 × 10^5^ DP cells. 4-Microtissues had a spheroidal structure. 5-Cell viability in DP microtissues on EVAL is much higher than that in DP spheroids generated by hanging drop method.	1-Self-assembly of DP cells into spheroidal inductive microtissues can be facilitated when cells are seeded at appropriate densities on EVAL surface. 2- DP microtissues mixed with newborn mouse epidermal cells and injected into the hypodermis of nude mice are able to induce new hair follicles (HFs).	[88]
		Human DP cells were isolated from scalp tissues obtained from plastic surgery. Human DPCs were expanded under conditions of activation of the Wnt/β-catenin signalling pathway (GSK-3 inhibitor, 6-bromoindirubin-3′-oxime (BIO)). hDP cells were analysed in the presence or absence of BIO. A cellular grafting assay to evaluate the hair-inducing ability of cultured human DP cells (5 × 10^6^ cells/grafting) was engrafted on the dorsal skins of nude mice (bulb/c, nu/nu)	1-Protein level of LEF1 in BIO-treated hDP cells showed a 2.7-fold increase compared with that without BIO. 2-Nuclear β-catenin was evidently observed in BIO-treated human DP cells 3-Human DP cells cultured in the presence of BIO and transplanted with murine epidermal cell fraction, formed hair follicle–like structures.	1-Human DP cells cultured under Wnt/β-catenin signalling activation by GSK-3β inhibition maintained the expression level of DP marker genes. 2-Human DP cells showed constant hair induction when transplanted with murine epidermal cell fraction.	[89]
Primary human interfollicular dermal fibroblasts, hair follicle dermal papilla, or hair follicle dermal sheath cells into rat tail collagen and human keratinocytes	In vitro and in vivo studies -SCID mouse	Cells were isolated from occipital scalp In vitro Constructs of collagen were established in parallel from fibroblasts, dermal papilla cells and dermal sheath cells. After 7 days, keratinocytes added and cultured for 17 days. In vivo Dermal fibroblasts, dermal papilla cells, dermal sheath cells and keratinocytes were trypsinized. 10^7^ million keratinocytes were then combined with either 10^7^ fibroblasts, or 10^7^ dermal papilla cells and pipetted into a hole located on a silicone chamber previously implanted under de dorsal skin of a SCID mouse. After one week the silicone chamber was removed, and cells were left for a further two weeks.	In vitro 1-There were no obvious differences in the equivalents containing hair follicle dermal cells, when compared to a fibroblast support layer. 2-Hair follicle dermal cells were capable of supporting growth and differentiation of overlying epidermal cells. 3-Type IV collagen (COL4) labeling was most intense in the basement membrane of constructs supported by dermal sheath cells from the hair follicle. In vivo 1-After grafting, all mixed cells were organized to form a skin structure. After 3 weeks, cells had reorganized with dermal cells on the inner surface, and keratinocytes exposed to the external air interface	1-Human hair follicle dermal cells can be readily interchanged with interfollicular fibroblasts and used as an alternative cell source for establishing the dermal component of engineered skin both in vitro and in vivo.	[90,91]
Human dermal papilla (DP) and dermal sheath (DS) cells and epithelial cells into collagen gel. Organotypic culture	In vitro and in vivo study -Nude mice	DP cells and DS cells of the human hair follicles (HFs) were isolated from human scalp. Dermal fibroblast (FB) from the scalp skin or foreskin was cultured separately as routine method. Keratinocytes from interfollicular and follicle outer root sheath (ORS superior or inferior) were also isolated from human scalp. DP cells, DS cells or FBs were embedded onto collagen gel and 5 × 10^5^ epithelial cells were plated on the prepared mesenchymal cell-populated collagen gels (Organotypic culture). Gels were immersed for 2 weeks and then were transplanted on the dorsal skin of the nude mice for 1 month.	1-Compared with the free cell-populated gels, on the four types of mesenchymal cell-populated gels, the growth, differentiation and shape of interfollicular keratinocytes and ORS epithelial cells were remarkably improved. 2-Epidermal structure reformed by the interfollicular keratinocytes on four types of mesenchymal cell-populated was differentiated very well and was similar to the epidermis of nature live skin. 3-Epidermis reformed by the superior ORS cells and the interfollicular keratinocytes was the thickest while the epidermis reformed only 3–4 layers by the bulb matrical cells was the thinnest. 4-In the organotypic culture of superior + DS and inferior + DS cells, the ORS cells reformed a long-shape hair follicle structure that was seen under microscope.	1-Dermal papilla cells induced superior and inferior epithelial cells to form hair follicle on nude mice. 2-Low passage dermal papilla cells mixed with hair follicle epithelial cells reformed many typical hair follicle structures and produced hair fibres after transplantation on nude mice. 3-Dermal part of hair follicle, such as dermal papilla cells and dermal sheath cells, has the ability to induce hair follicle formation by interaction with the epithelial cells of hair follicle.	[92]
Human dermal papilla (DP) cells, hair follicle enriched primary cultures (HFSCs) and immortalized human bulge stem cell line Tel-E6E7 and dermal fibroblasts (DF) on acellular dermal matrices Tissue engineered skin	In vitro and in vivo study -BALB/C nude mice	-DP cells and DF cells were isolated from occipital human scalp and HFSC cells were isolated from human skin biopsies. -Immortalized human bulge stem cell line Tel-E6E7 was acquired. -Porcine acellular dermal matrices (ADMs), were seeded with 5 × 10^5^ cm^2^ DF or DP cells as the dermal cellular component. They were cultured for 7 days. After that, matrices were inverted and seeded on the opposite side with 5 × 10^5^ cm^2^ HFSCs or Tel-E6E7 as the epidermal cellular and cultured for 21 days (combining liquid and air-liquid phases). -Tissue engineered skins were grafted int the back of BALB/C nude mice aged 8 weeks (*n* = 6 per group): one type of construct was seeded with HFSCs and DPCs (HFSC-DPC), a second type was seeded with HFSCs and DFs (HFSC-DF), and a third type was seeded exclusively with HFSCs without the dermal component. -Grafts were analysed at 14, 30 and 70 days.	1-Air-liquid interphase improved in vitro skin constructs: skin constructs with HFSCs alone or with DFs (HFSC-DF) showed an epidermis with a proliferative basal layer, incipient and irregular stratum spinosum, frequent dyskeratosis, and a cornified layer. In the HFSC-DPC skin constructs, most regular epidermis was observed. 2-In vitro skin constructs with DPCs showed a higher number of p63-Positive epidermal basal cells and epidermal invaginations. 3-Presence of DPCs favoured the graft-take of composite skin and improved the wound healing process. 4-After 14 days of grafting, the HFSC-DPC constructs showed the highest number of blood neo-vessels. Amount of VEGF was six fold higher than the amount in the DF cultures (927 ± 87 pg/mL vs. 147 ± 54 pg/mL). 5-Only the HFSC-DPC constructs grafted in the nude mice showed notable epithelial cyst-like inclusions in the remodelling dermis and expression of k6hf.	1-Presence of DP cells in composite skin constructs generated in air-liquid interphase led to the formation of an epidermal-like structure with the most regular stratification, more invaginations that could indicate hair follicle neogenesis attempts, and maintenance of an epidermal stem cell pool.	[93]
Human keratinocytes, fibroblasts, dermal papilla cells and GFP-tagged HUVECs. Human skin constructs (HSCs) in type I collagen gel.	In vitro and in vivo study -Male immunodeficient nude mice (athymic nude, Crl:NU(NCr)-Foxn1nu	-Neonatal dermal keratinocytes and fibroblasts were isolated from human foreskin. -GFP-tagged HUVECs were also cultured. -Dermal Papilla (DP) cells discarded scalp tissues from hair restoration surgery. -The molds with varying hair densities (19, 81, 255 hair follicles (HFs) per cm^2^) were 3D-printed. -3D skin constructs were generated with a type I collagen matrix as dermal compartment. 1.25 × 10^5^ fibroblasts/mL were added and polymerized around the 3D-printed HF molds. -After complete polymerization, the molds were removed and 3000 DP cells per microwell were added on top of the gel. The constructs were cultured overnight and then, 10^6^ keratinocytes (KCs) were added on top of the gel. -Constructs were cultured for 1–3 weeks. -GFP-tagged HUVECs were also studied in collagen type I gel at a concentration of 2 × 10^6^ cells/mL. -Human skin constructs were grafted onto male immunodeficient nude mice (athymic nude, Crl:NU(NCr)-Foxn1nu. They were maintained 5 days into a silicone chamber and then maintained for 4–6 weeks for in vivo analysis.	1-3D-printed technology allowed for precise control of DP cells aggregate size by adjusting the diameter of the microwells. The spontaneous aggregate formation restored the expression of versican (VCAN) and alkaline phosphatase (ALP) activity and suppressed the expression of smooth muscle actin (SMA). 2-Keratinocytes (KCs) over the dermal constructs and allowed the cells to settle down resembled a hair follicle-like unit (HFU) 3-Culture period of 3 weeks led to the elongation of the HFs down into the dermis and a better organization of the inner and outer root sheath layers. 4-Transfection of *Lef-1* in cultured DP cells, and later formation of spheroids restored the intact DP cells gene signature. Skin constructs generated with them resulted in significant increase (up to 13-fold) in the expression of the specific hair lineage genes, including the outer and inner root sheath (K17, K71, K25) and hair companion and medulla markers (K75). 5-To generate a vascular bed, GFP-tagged human umbilical vein endothelial cells (HUVECs) in the dermis of HSCs were included together with the dermal fibroblasts. 6-Immunofluorescent wholemount imaging of the constructs revealed that these capillary-like structures were near to the HFs. 7- Grafting the vascularized HSCs onto mice promoted host vascularization into the grafts 8-Four to five weeks after grafting, vascularized HSCs presented substantial hair growth in the grafts, whereas the HSCs prepared with FB aggregates did not induce hair formation.	1-This tissue engineering exploits the epidermal-mesenchymal interactions during hair development while synthetically guiding the physiological conformation and reconstituting the gene signature of cultured cells to induce human hair growth in vitro and in mice.	[94]

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
