# Peer review of "Advanced Medical Therapies in the Management of Non-Scarring Alopecia: Areata and Androgenic Alopecia"

_ijms, 2020, doi:10.3390/ijms21218390_

Round 1
Reviewer 1 Report
Dear Authors,
This is an ambitious comprehensive review about the cell therapy (including genes, cells, and tissue engineering) for alopecia.
Although the paper is relatively well written, one question is needed to be concerned. Alopecia is a description about pathological hair loss in clinical setting. Thus, it is a large disease entity, including numerous diseases caused by different pathophysiology, such as hereditary factor (androgenetic alopecia), autoimmune factor (alopecia areata and frontal fibrosing fibrosis), infection / trauma (scarring alopecia), autoinflammation and aging process.
It is impossible to use a single article to review this broad field. The title, “Advanced medical therapies in the management of alopecia” may not be appropriate for covering such as a wide range of disease. Do the authors prefer to narrow the scope of this article in a more specific disease or field.
On the other hand, the structures of tables are scattered and not well-organized. It needs a revision.
Author Response
Dear Reviewer 1,
Thank you for your suggestions and comments. We would like to thank you for the identification of areas of our manuscript that needed modification and also for your great score.
Please find enclosed a revised manuscript, which we have prepared in accordance with your comments for its improvement before publishing. We have narrowed the scope of this paper to the most frequent forms of non-scarring alopecia: alopecia areata and androgenic alopecia. We have also changed the title of the article. Moreover, we have changed the structure of the tables. We had a problem during the PDF building and now we have submitted the modified tables in a different document.Reviewer 2 Report
General comments
1. This is a comprehensive review article about advanced medical therapies in treating hair loss.
2. Table contents are not well organized. It is not easy to read in current form. Please arrange the contents in better way.
3. There is a treatment called Rigenera, which utilized progenitor cell rich micrografts to enhance hair growth. There is a paper published this year. I think it is better to add this treatment in this review article.
4. About tissue engineering for alopecia, there are several ongoing clinical trials. Please add these data in this paper.
5. Please address more potential side effects and contraindications of these new treatments.
Specific comments
- page 6, line 5, add bacterial and fungal overgrowth.
- page 6, line 14. Immunoreactivity of CD15...present in AHA. Please explain this sentence more detail.
- page 12, the line of table, confirmed diagnostic of hair loss. Please define which type of hair loss.
- page 16, line 18, double in
- page 26 and 27, Reference's typesetting should be re-arranged.
- page 30, line feed between two references
- page 33, Reference's typesetting should be re-arranged.
Author Response
Dear Reviewer 2,
Thank you for your suggestions and comments. We would like to thank you again for the identification of areas of our manuscript that needed modification and also for your great score.
Please find enclosed a revised manuscript, which we have prepared in accordance with the reviewers’ comments for its improvement before publishing. Moreover, we have included below the relevant changes made according to your comments: General comments: -We have re-arranged the tables in order to simplify the information. We had a problem during the PDF building process and now we have submitted the tables in a different document. -We have included information about Rigenera in the manuscript. -We have added ongoing clinical trials in tissue engineering section. -We have included the potential side effects of mesenchymal stem cell treatments. Specific comments: -Bacterial and fungal overgrowth have been added. -Immunoreactivity of CD15 has been explained. -In the table, hair loss has been defined. -Reference's typesetting and spaces have been re-arranged.Round 2
Reviewer 1 Report
I have no further major question about this article, except there are some typos:
Line 99: Procurement and production of mesenchymal -> Procurement and production of mesenchymal stem cells
Line 185: demoscopic -> dermoscopic